# ASIF: Coupled Data Turns Unimodal Models to Multimodal without Training

**Antonio Norelli** *

**Marco Fumero**\*   **Valentino Maiorca**\*   **Luca Moschella**\*

**Emanuele Rodolà**\*   **Francesco Locatello**†

\*Sapienza Università di Roma, dipartimento di Informatica

†Institute of Science and Technology Austria (ISTA)

## Abstract

CLIP [1] proved that aligning visual and language spaces is key to solving many vision tasks without explicit training, but required to train image and text encoders from scratch on a huge dataset. LiT [2] improved this by only training the text encoder and using a pre-trained vision network. In this paper, we show that a common space can be created without any training at all, using single-domain encoders (trained with or without supervision) and a much smaller amount of image-text pairs. Furthermore, our model has unique properties. Most notably, deploying a new version with updated training samples can be done in a matter of seconds. Additionally, the representations in the common space are easily interpretable as every dimension corresponds to the similarity of the input to a unique image-text pair in the multimodal dataset. Experiments on standard zero-shot visual benchmarks demonstrate the typical transfer ability of image-text models. Overall, our method represents a simple yet surprisingly strong baseline for foundation multimodal models, raising important questions on their data efficiency and on the role of retrieval in machine learning.

## 1   Introduction

Large multimodal models such as CLIP [1] are rapidly becoming the standard for foundation models [3] in computer vision. This is largely due to their zero-shot and open-world capabilities that enable diverse suites of downstream tasks, from classification to detection and visual search.

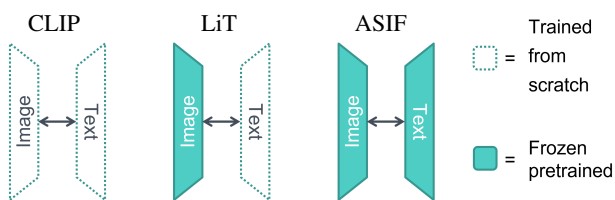

Figure 1: ASIF is a simple recipe to align the representations of two frozen pre-trained encoders.

Overall, Radford et al. [1] demonstrated that treating image recognition as language interpretation allows generalizing to a multitude of tasks without training explicitly for them. By building an interpreter, CLIP changed the way computer vision is approached [4]: rather than extracting the "dog" label from an image like a CNN [5], CLIP tests the hypothesis of a dog being in the image against all the other hypotheses. The success of this image-text association is a testament to the power of deep learning: the CLIP model was the first of its kind, and building it required a joint training of two large neural encoders on a vast collection of image-text pairs.

Correspondence to Antonio Norelli <norelli@di.uniroma1.it>. A demo sufficient to reproduce the main results in the paper within minutes, even from smartphone, can be found here: `https://github.com/noranta4/ASIF`

37th Conference on Neural Information Processing Systems (NeurIPS 2023).

Still, training neural networks at such scale presents several challenges beside the obvious infrastructure and training costs. Notably, it requires collecting massive training sets, making it difficult to interpret the predictions of the model in light of their training data. Additionally, the training assets are often not owned by the institution training the model [6].

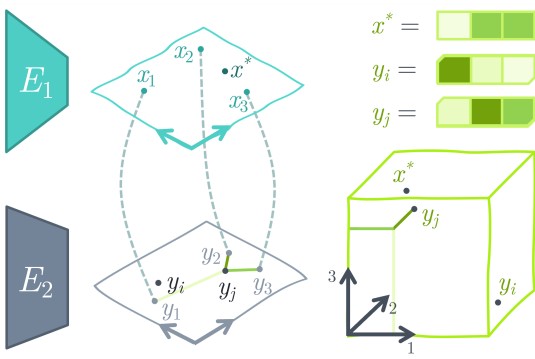

Figure 2: **The ASIF construction.** An ASIF model is defined by two unimodal pretrained encoders and a collection of coupled embeddings. This is sufficient to compare elements from different modes through representations made of similarities with ground-truth pairs: $y_j$ is more similar to $x^*$ than $y_i$.

This introduces several additional challenges, from reproducibility to the difficulty of ensuring that an asset owner can remove their data from the model [7–11]. Overall, these considerations make large multimodal models relatively inaccessible to researchers and practitioners until checkpoints are released or access to demo is granted. And even then, the ability to tweak the models by adding or removing training data or to interpret their results is limited.

In this paper, we present ASIF, a simple non-parametric procedure that turns pretrained uni-modal image and text encoders into a multimodal model using a *relatively small*[1] collection of image-text pairs and no additional training, as depicted in Figure 1. The resulting model is functionally equivalent to CLIP, effectively producing aligned representations of images and their captions.

**Intuition.** The key insight is that captions of similar images are themselves similar (Fig. 3), and therefore a representation crafted using just similarities to ground-truth multimodal pairs is quasi mode-invariant (Fig. 2).

The ASIF procedure is not only efficient but also has several intriguing features built-in. One of the key advantages is the ability to easily edit the model - adding new image-text pairs or removing outdated ones is as simple as encoding or canceling the corresponding embeddings. Furthermore, the multimodal representations are highly interpretable, as each dimension corresponds to the similarity of the input to a specific entry in the multimodal dataset.

**Contribution.** Our results are surprising and raise several questions. Despite (1) the simplicity of the approach, (2) a multimodal dataset that is up to 250 times smaller than in prior work and (3) the lack of actually training the model on multimodal data, ASIF achieves zero-shot classification accuracy on downstream datasets that is comparable to CLIP [1, 2]. This raises important questions on the data efficiency in foundation models, making ASIF a very powerful and cheap baseline for future work, and opening new doors for data centric AI [12].

In summary, we:

- Introduce the ASIF procedure, which turns two pretrained unimodal black-box encoders into an interpretable multimodal model without tuning a neuron.

- Demonstrate the effectiveness of ASIF models on zero-shot image classification tasks, where they achieve performance in the same ballpark of CLIP with significantly fewer image-text pairs.

- Discuss the unique properties of ASIF, its implications on the role of memory and retrieval in machine learning, and the new opportunities it opens.

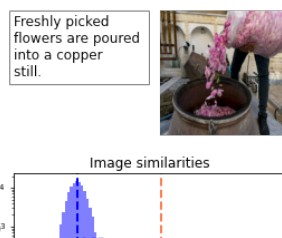
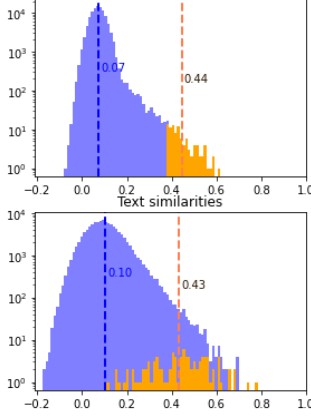

Figure 3: **Captions of similar images are themselves similar.** Distribution of similarities of 100k embedded pairs in the training set versus the above image and caption embeddings. We highlighted in orange the 1000 pairs with the highest image similarity.

---

[1] CLIP [1] experiments used from 400M to 15M captioned images as training samples, LiT [2] from 901M to 10M. Our experiments use 1.6M.

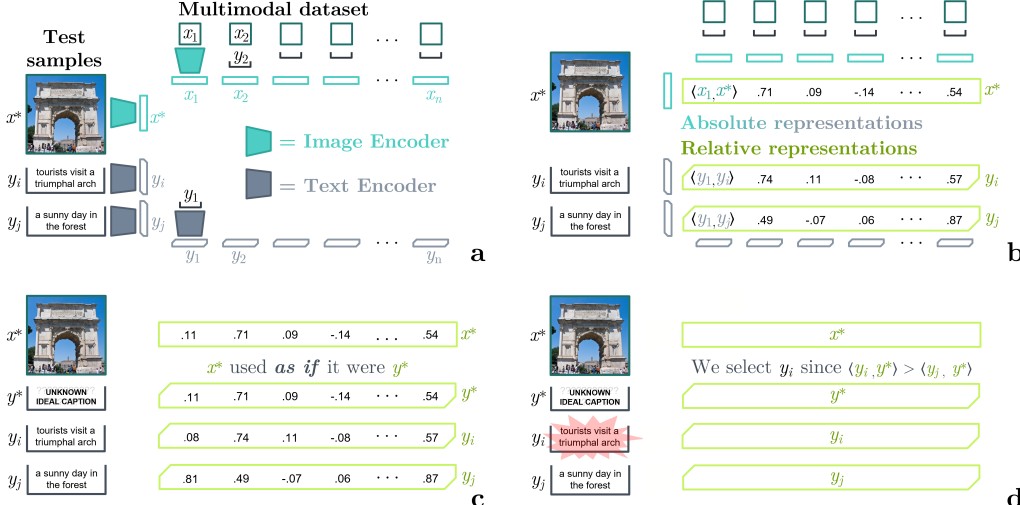

Figure 4: **Zero shot classification with ASIF.** In this example we determine the best caption for the image $x^*$ from the two candidates, $y_i$ and $y_j$. **a.** Compute and store the embeddings of the multimodal dataset and the test samples. **b.** Compute the relative representation of the test image and the candidate captions. **c.** We consider the relative representation of $x^*$ with respect to the image collection $x_1, \ldots, x_n$ *as if* it was the relative representation of $y^*$ – the ideal caption for $x^*$ – with respect to the corresponding caption collection $y_1, \ldots, y_n$. **d.** We choose the candidate caption most similar to the ideal one.

## 2 Aligning Pre-Trained Models with ASIF

In the following we present how a collection of captioned pictures implicitly defines a common space for images and texts through relative representations [13], allowing to build a multimodal model without training. Here we focus exclusively on vision and language as modalities due to the wider availability of relevant models and paired data. However, we anticipate that our procedure could be more generally applicable. Indeed, subsequent research by other teams has already utilized ASIF as a baseline in other modalities, such as audio [14]. Before delving into the specifics of our method, we will briefly review existing techniques for establishing this common space.

**Contrastive training to build a common space.** With multimodal models, we refer to architectures that embed inputs of diverse modes into the same space. The better is a multimodal model, the closer are representations of different modes of the same object. So far, this common space has been obtained as the result of a contrastive training of one [2] or both the neural mode encoders [1, 15]. Using a collection of image-text pairs as training set, a contrastive loss promotes the closeness of image and text embeddings from the same pair, while spacing out the ones from distinct pairs. Zhai et al. [2] train just the text encoder to match the image embeddings of a pretrained visual encoder. Once the encoders are trained, a multimodal model can be adapted to perform any visual classification task just by crafting a caption for each label and selecting the one with the embedding closer to the embedding of the input image (zero-shot image classification).

**Relative representations.** Our idea to build a common latent space is to use a collection of coupled data as a "rosetta stone" between modalities, and represent each new data point as its similarities to the points of the same modality in the collection. In other words, we compute a *relative representation* for each new data point:

**Definition 2.1.** Given an encoder $E : X \to \mathbb{R}^d$ and a subset of samples $\{x_1, \ldots, x_n\}$ denoted as anchors, we define the relative representation of $x'$ as the $n$-dimensional vector:

$$x' = [\text{sim}(x', x_1), \ \ldots \ , \text{sim}(x', x_n)]$$

for some similarity function sim, e.g. the cosine similarity. $x_i \in X$ are input samples, e.g. images or texts; $x_i \in \mathbb{R}^d$ are the embeddings of $x_i$ obtained with the encoder $E$, i.e. the absolute representations of $x_i$; while $x_i \in \mathbb{R}^n$ are the relative representations of $x_i$.

We observe that when each anchor is available in two or more modalities, we can compute relative representations of samples from those modalities using the same subset of anchors. Most notably, these relative representations will all live in the same space, even when they are representing samples from different modalities. This foundational insight is illustrated in Figure 2.

**Relation with Kernel methods.** Definition 2.1 may not look surprising to readers familiar with the literature on kernel methods [16]. Instead of presenting $x'$ as a kernel, we say it is a relative representation to stress that (1) we want to *explicitly* represent the coordinates in our ASIF procedure as opposed to operating in an implicit feature space and (2) we do not aim at learning regression parameters, although we note this may help with the inference speed. Instead, we rely on a simpler procedure that may be interpreted as a hard version of the Watson-Nadaraya [17, 18] regression with a distance threshold. Nevertheless, it is worth noting that while the ASIF procedure hinges on the ability to compute similarities between samples, such computation can be achieved using a kernel function, thus sidestepping the need for explicit representations generated by unimodal encoders. Although integrating kernel methods could offer benefits, as alluded to earlier, delving into these prospects is beyond the scope of this paper. Our central focus remains on illustrating how single-domain pre-trained networks can be aligned without additional training.

**ASIF: relative representations inducing a meaningful common space.** Consider the embedding spaces of any two good text and visual encoders, we expect captions of images that are close in the visual space to be themselves close in the language space. This fact makes a representation defined in terms of similarities against a set of ground-truth multimodal pairs almost mode-invariant, i.e. an image and its caption share almost the same representation.

That is why we can assign the best caption to a new image $x^*$ just by performing nearest neighbors in this new space: we can consider the relative representation of $x^*$ respect to the image collection $(x_1, \ldots, x_n)$ *as if* it was the relative representation of its ideal caption $y^*$ with respect to the counterpart collection $(y_1, \ldots, y_n)$, see Figure 4. The whole procedure to set up an ASIF model and use it to find the best caption for a new image follows.

**ASIF recipe.** Ingredients:

- Two good encoders, each mapping a single data modality to a vector space. Let $X$ and $Y$ be the mode domains, for instance a pixel space and a text space, we need $E_1 : X \rightarrow \mathbb{R}^{d1}$ and $E_2 : Y \rightarrow \mathbb{R}^{d2}$.

- A collection of ground truth multimodal pairs: $D = \{(x_1, y_1), \ldots, (x_n, y_n)\}$, for instance captioned images.

Procedure to find the best caption among a set of original ones $\hat{Y} = \{\hat{y}_1, \ldots, \hat{y}_c\}$ for a new image $x^*$:

1. Compute and store the embeddings of the multimodal dataset $D$ with the encoders $E_1, E_2$ and discard $D$. Now in memory there should be just $D_E = \{(x_1, y_1), \ldots, (x_n, y_n)\}$;

2. Compute the $n$-dimensional relative representation for each candidate caption $\hat{y}_i = [\text{sim}(\hat{y}_i, y_1), \ldots, \text{sim}(\hat{y}_i, y_n)]$, where sim is a similarity function, e.g. cosine similarity. Then for each $\hat{y}_i$ set to zero all dimensions except for the highest $k$, and raise them to $p \geq 1$. Finally normalize and store the processed $c$ vectors $\hat{y}_i$. Choose $k$ and $p$ to taste, in our experiments $k = 800$ and $p = 8$;

3. Compute the relative representation of $x^*$ using the other half of the embedded multimodal dataset $D_E$ and repeat the same processing with the chosen $k$ and $p$;

4. We consider the relative representation of the new image $x^*$ *as if* it was the relative representation of its ideal caption $y^*$, i.e. we define $y^* := x^*$. So we choose the candidate caption $\hat{y}_i$ most similar to the ideal one, with $i = \text{argmax}_i(\text{sim}(y^*, \hat{y}_i))$.

To assign one of the captions to a different image $x^{**}$ repeat from step 3.

**Properties of ASIF models.** The above construction yields several intriguing properties for free:

*No training and "data centric".* As we have seen, an ASIF model is built on top of two independently pretrained encoders and the embeddings of a multimodal dataset, and so without training or finetuning any neuron. Being deployable or updatable in seconds, an ASIF model is radically "data centric" [12]. For example, it is trivial to adjust the model by adding or forgetting specific samples. The latter use-case is particularly important, as the right to use specific assets may change over time and removing the effect of specific samples from a trained network requires sophisticated forgetting techniques, e.g. [7–11]. In our procedure, the encoders should be pre-trained with established data sets that do not change over time, while removing the effect of a multimodal pair is as simple as deleting its embeddings.

*Data efficiency:* Being able to exploit two pretrained encoders, ASIF models require far less ground-truth multimodal pairs to become effective. As confirmed by our experiments, ASIF reaches competitive zero-shot performance on diverse classification benchmarks by using a fraction of the multimodal data of its predecessors, reaching a respectable accuracy even with thousands of pairs (we refer to Section 3 for more details). This is in line with classical work in computer vision, where prototypical networks [19] are a strong baseline in the extremely few-shot regime.

*Interpretability:* Sparse relative representations make the ASIF models interpretable classifiers. In fact, we can trace back every prediction to a small set of data points in the multimodal dataset –corresponding to the dimensions that are nonzero in both the image and the label relative representations– accountable for the outcome (at most $k$), see Figure 6. This enables visualizations of the relevant samples contributing to a correct or incorrect prediction at no extra cost, in stark contrast with other approaches that are often costly [20] and fragile [21, 22].

**Relation to k-Nearest Neighbors.** Like k-NN, ASIF is a non-parametric supervised learning algorithm that requires an explicit representation of every entry in the training dataset at test time. Differently from k-NN, ASIF can perform open-ended classification, as shown e.g. in Figure 4 with two competing brand new captions. Indeed, ASIF is functionally equivalent to CLIP, and can function as a drop-in replacement in applications using CLIP-like models.

**Implementation that scales.** Clearly, our method pays the price of avoiding training with a larger memory footprint and increased computation at inference time, since we need to compute not only the embeddings but also the cosine similarities against the multimodal dataset. As such, our approach should not be considered a general one-stop replacement for CLIP, although in our experiments we managed to scale ASIF to 1.6M pairs while maintaining a reasonable inference speed. Our non-optimized implementation of ASIF is less than 2x slower than CLIP. On a positive note, there are two well-established techniques that could radically enhance the efficiency of ASIF and potentially enable scalability to billions of entries. The memory footprint required to store all the embeddings of the multimodal dataset can be dramatically reduced using product quantization [23], while inverse indexing [24] can be used to circumvent the need for computing the cosine similarities against the entire dataset at test time. These techniques are both implemented e.g. in the FAISS library [25]. Finally, we find that the distribution of pairs chosen during inference is rather short-tailed, presenting opportunities to massively prune the model even *at deployment time*, deleting from memory the data that is never used. It should be noted, however, that the assessment of the performance of large-scale optimized ASIF models is beyond the scope of this work. While it is an interesting direction for future research, the focus of our current study is on establishing the potential the ASIF method.

## 2.1   Design choices and implementation of ASIF models.

**Curating the multimodal dataset.** While neural models like CLIP or LiT are defined just by the weights of the two encoders, to univocally identify an ASIF model we also need the embeddings of the multimodal dataset. Even if two image-text ASIF models rely on the very same encoders, they comply to different visual understandings of the world if they are based on different collections of image-text pairs, therefore achieving different performances on the same tasks. Unlike conventional neural vision models, ASIF enables effective curation of the training dataset through swift iterations, given the absence of training and the smaller datasets. Furthermore, ASIF provides the means to assess the impact of each training pair, as demonstrated in Figure 6.

**Salient hyperparameters.** While using the raw relative representations already produces an ASIF multimodal model with non-trivial capabilities, we found that two simple treatments greatly improve performance and efficiency, and also foster interpretability.

| Method | Dataset size | ImageNet | CIFAR100 | Pets | ImageNet v2 |
|---|---|---|---|---|---|
| CLIP [1] | 400M (private) | 68.6 | 68.7 | 88.9 | - |
| CLIP [1] | 15M (public) | 31.3 | - | - | - |
| LiT [2] | 10M (public) | 66.9 | - | - | - |
| CLIP( Zhai et al. 2, uu) | 901M (private) | 50.6 | 47.9 | 70.3 | 43.3 |
| LiT [2] | 901M (private) | 70.1 | 70.9 | 88.1 | 61.7 |
| ASIF (sup vis. encoder) | 1.6M (public) | 60.9* | 50.2 | 81.5 | 52.2 |
| ASIF (unsup vis. encoder) | 1.6M (public) | 53.0* | 46.5 | 74.7 | 45.9 |

Table 1: **Zero shot classification accuracy of different multimodal designs.** CLIP and LiT implementations vary by dataset and the visual transformer used as image encoder. The first CLIP and LiT entries use a VITb16 as ASIF, the last CLIP and LiT entries use a VITb32 (larger patch size). The public dataset of CLIP is a curated subset of YFCC100m [38], while LiT and ASIF use CC12M.
*We used a subset of the ImageNet validation set to tune the two hyperparameters of ASIF which were then used on the other data sets. The number reported in the table is a test set. When tuning on different datasets, accuracies stay consistent, see the appendix.

i) *Sparsification.* We set to 0 all the entries of the $n$-dimensional relative representation except for the top $k$. In our experiments $n$ and $k$ are respectively in the order of millions and hundreds. In this way we cut off the small noisy signals from the dissimilar entries, that accumulated during comparisons would destroy the signal from the few similar entries. Furthermore we get highly interpretable representations that can be efficiently compared, since we have just $k$ nonzero features, each one linked to a single entry in the multimodal dataset.

ii) *Exponentiation.* We raise all the nonzeroed similarities $sim(x', x_i)$ to $p$, with $p \geq 1$. This non-linearity weighs more the contribution of the most similar entries in the relative representation.

Besides the pivotal choice of the ground-truth multimodal pairs, the number of non-zero elements $k$ and the exponent $p$ are the salient hyperparameters to consider when deploying an ASIF model. In general, we found that picking a $p \neq 1$ may help, while choosing a $k \ll n$ is always crucial. For more details see Section 3.

## 2.2 Closely Related Works

**Classics.** In [26], Stanley Ulam affirms that a mathematical formalization of the word "as"–on a par with the connectives "and", "or", "implies" and "not"–would be a key milestone to artificial intelligence. This idea of analogies being at the core of cognition is shared by Hofstadter [27], who states that a concept is a collection of analogies, in line with what the ASIF procedure prescribes.

**Retrieval augmented foundation models.** Recent works in NLP enhance unimodal language models with retrieval to reduce the size of the architectures and the training dataset while also making results more transparent [28, 29]. Our work is in line with this view, that the ASIF procedure extends to the multimodal case. Importantly, ASIF offers a new perspective on data centric AI [12], where data and the external memory implement the alignment between modalities. Networks with discrete Key-Value bottlenecks [30] are closely related to our approach, with the critical differences that our memory is not learned and that their decoder is trained for classification. Retrieval and memory-augmented models have also been successful in Reinforcement Learning [31], physical reasoning [32], and code generation [33]. Finally, we notice that making predictions on new samples by exploiting the similarity with a dictionary of previous data points is a common approach in computer vision [19] for few-shot learning. Our procedure is also related to compressed sensing algorithms where a signal (in our case an image) is sensed as a sparse combination of fixed atoms [34, 35] with an iterative projection procedure [36, 37] and only transmitting the coefficients to the receiver (text modality).

**Learning multimodal models.** Following the intuition outlined by early works on aligning text and image embeddings [39, 40], today large multimodal models are conquering the computer vision scene thanks to their wide applicability and easy transfer to new downstream tasks [1, 2, 15, 41].
We identify two key leaps respect to traditional models like ResNet [42]: (i) Free text allows to learn visual concepts beyond a finite set of predefined categories and to exploit the structure of language to compose them, as masterfully seen in Dall-E [43]. (ii) The recognition tag transitioned from being an output pulled out of the image by the neural stack (the label) to become an input that should be interpreted, and therefore processed by its own encoder (the free text caption). This corresponds to an epistemological perspective shift, as discussed by Norelli et al. [4]. Data and learning efficiency are clear challenges for large multimodal models, that often require hundreds of millions of examples. Efforts such as [2, 29] attempt to reduce this. ASIF presents a different take on this problem, showing how much can be achieved by simply remembering the training data efficiently.

# 3 Empirical Evidence

In the following we compare ASIF to traditional multimodal models based on contrastive training, CLIP and LiT. We then take a closer look to the classification of a single image, unpacking the relative representations and following the classification algorithm step by step. As a prelude to the above, we start by discussing the pretrained encoders and dataset forming the ASIF models we tested.

**Pretrained encoders and multimodal dataset used.** For our primary experiment, we utilized vision transformers as image encoders, pretrained either in a supervised manner (DEIT base, [44]) or in an unsupervised manner (DINO ViTs8, [45]), on Imagenet 1k [46] and 21k [47] respectively. The embedding size was 768 for DEIT and 384 for DINO. This configuration aligns with that used in LiT [2], with the sole distinction being that, unlike LiT, we used a pre-trained, frozen text encoder. Regarding the text encoder, we employed the SentenceT transformer [48], trained on a dataset comprising more than 1 billion sentences obtained from the internet. We employed the first 1.6M entries of the Conceptual Caption dataset (CC12M, [49]) as our multimodal dataset. This dataset amasses images and filtered alt-texts collected from the internet. To optimize performance on a single Tesla T4 GPU, we limited our analysis to the initial 1.6M pairs. In our "scaling-laws" experiments, we also utilized DEIT tiny and small vision transformers [44] and two smaller SentenceT encoders.

**Zero-shot performance.** We assessed the quality of our ASIF multimodal model by comparing its zero-shot classification performance against CLIP and LiT on four datasets: CIFAR100, Imagenet, Imagenetv2, and PETS [46, 50–52]; see Table 1. We crafted label prompts as in Zhai et al. [2, Table 11].

Remarkably, we achieve competitive performance with CLIP and LiT using two frozen pretrained encoders and a fraction of the image-text pairs.

The two ASIF models reported in Table 1 differ for the vision encoder, that is respectively supervisedly (DEIT) and unsupervisedly pretrained (DINO). We tuned $k$ and $p$ on the ImageNet validation set, in both cases we used $k = 800$ and $p = 8$. The comparison between the two versions is not completely fair since the visual transformer architecture of DINO is smaller (e.g. the codings are 384-dimensional rather than 768) but corroborates the effectiveness of ASIF with encoders pretrained using no supervision. In the appendix we report the results of a wider collection of ASIF models based on different visual and textual backbones.

*Summary:* Overall, we observe that our ASIF procedure can achieve competitive zero-shot results with a fraction of the image-text pairs used in prior work (assuming access to other pre-trained, even unsupervisedly, unimodal models).

**ASIF scaling laws.** In Figure 5 we show the full zero-shot accuracy trend on Imagenet as the size of the multimodal dataset grows for different choices of $k$ and $p$. ASIF models become effective very quickly: we reach a non-trivial $18\%$ classification accuracy using just the first 10,000 pairs in the multimodal dataset. We recall that building an ASIF model from scratch still requires a lot of unimodal data to pretrain the encoders, but now this data may come untied and even without any label.

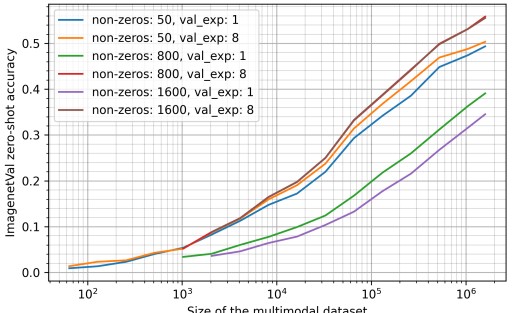

Figure 5: **ASIF is a learning algorithm:** Imagenet accuracy improves smoothly as we increase the size of the multimodal dataset. Colors show the impact of $k$ and $p$ (non-zeros, val exp).

We tested ASIF further with smaller image and text encoders on multimodal datasets of different sizes ($10^2$ to $10^6$ image-text pairs from CC12M) to provide early evidence about ASIF scaling laws. We used DEIT tiny, small, and base vision transformers [44], along with two smaller SentenceT encoders. As we see in Figure 5 and in the appendix, Imagenet classification accuracy grows smoothly with the size of the multimodal dataset, and performance does not saturate earlier with smaller encoders. These results are promising but still preliminary, further experiments with larger multimodal datasets are left for future work.

*Summary:* Our experiments show a steady improvement in ASIF's performance as the size of the multimodal dataset increases. Performance deteriorates with smaller encoders, but even then there is no sign of saturation or plateau.

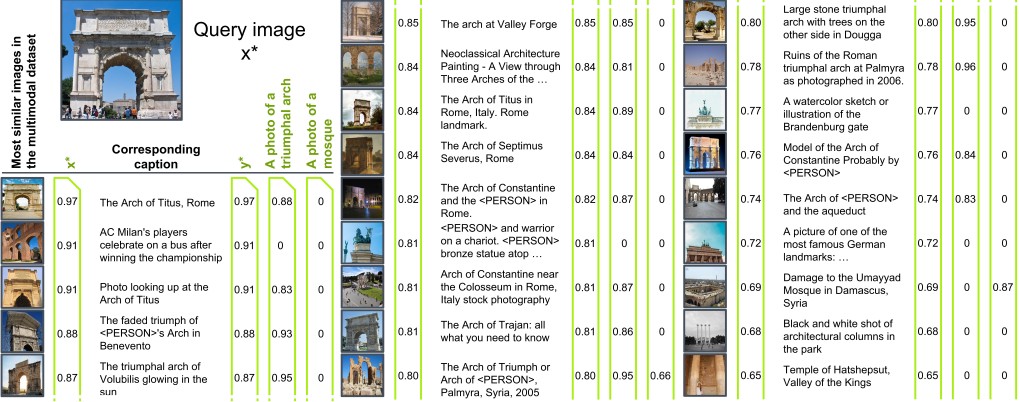

Figure 6: **ASIF representations are interpretable and amendable.** Thorough analysis of the relative representations –the four vectors in green– produced by ASIF to assign the best caption to a given image $x^*$. Every dimension corresponds to the similarity of the input image (text) to a unique image (text) entry in the multimodal dataset. We can visualize the training samples leading to correct or incorrect predictions, which is useful to curate better data. For example, the second training sample is broken, we can remove it and produce an updated ASIF model in seconds.

**Adjusting an ASIF model in seconds.** Consider classifying EuroSAT [53] satellite images using ASIF. Initially, the zero-shot performance outperforms random chance but is not impressive ($29.4\%$ unsup. configuration, 10 classes). CLIP, while better, also falls short with a $54.1\%$ performance rate.

Now, imagine we acquire 100 new image-text pairs from EuroSAT. Our goal is to develop a new multimodal model based on an updated training set, anticipating the need to process more satellite images in the future. With CLIP, this would require us to fine-tune or retrain the model from scratch using the updated dataset. In contrast, ASIF requires only to obtain and store the new pairs' embeddings. The model retains its usability for all the previous images, but it now also makes accurate predictions for satellite images, achieving an $82.2\% \pm 2.0$ accuracy on EuroSAT.[2]

Similarly, if we need to remove samples from the multimodal training dataset—either because they are faulty (as shown in Figure 6) or because we have lost the license to use them—the process is as simple as deleting the corresponding embeddings to get a new model.

*Summary:* ASIF enables quick model adjustments and data handling without the need for retraining, unlike traditional models such as CLIP. This demonstrates its practicality in real-world scenarios, such as the emergence of a new setting or the loss of rights for assets used during training.

**Deep dive into a classification.** To better understand why and how the ASIF procedure works, we are going to follow step by step the single classification depicted in Figure 6, showing the entries of the multimodal dataset used to build the relative representations. For simplicity we assume $k = 23$.

We want to find the Imagenet label of the upper left image in Figure 6. The first step is to compute its relative representation with respect to the multimodal dataset, this procedure selects the 23 most similar images in our collection. The most similar are indeed triumphal archs that–as similarity decreases–turn to generic structures with archs or ancient monuments. No other image in the multimodal dataset is involved in the classification, we could replace all the entries in our collection except for these 23, and as long as no new image is more similar to the input image, every computation in the classification process would remain the same. This consistency of the ASIF procedure is in contrast to traditional approaches like CLIP and LiT, where the parameters defining the encoder computations are shaped by the whole multimodal dataset through contrastive learning.

Now, we approach the pivotal step of the ASIF procedure: jumping into the text space by treating the relative representation of the input image *as if* it were the relative representation of its ideal caption. The vector remains the same, but its meaning changed: now it signifies how much the ideal caption should be similar to 23 captions.

---

[2]Why does CLIP perform better zero-shot? Likely, its 400M pairs have more samples close to satellite images than ASIF's 1.6M samples from CC12M, evidenced by ASIF's drastic improvement with just 100 new samples.

The efficacy of this jump hinges entirely on the quality of the multimodal dataset. Looking at the 23 captions in Figure 6, we are confident that the most fitting candidate caption corresponds to one of the prompts associated with the `triumphal_arch` label in ImageNet, such as *"a photo of a triumphal arch"*. Conversely, a dataset featuring 23 non-informative captions, such as file names *"IMG_20180823.jpg"* or camera settings *"D90 18.0-70.0 mm f/3.5-4.5"*, would not allow the model to recognize the correct class even with the best image and text encoders. This limitation is actually a defining feature of the ASIF procedure, as the meaning attributed to the input is ultimately determined by the multimodal dataset and not by the encoders: by acting just on the image-text pairs, we have full control over the model's output.

*Summary:* Our simple (non-cherry picked) example showcases how the ASIF predictions can be easily attributed to specific examples in the multimodal training data by construction. This feedback can be used to explain predictions and grow high quality datasets in data centric AI, for example by inspecting which examples contribute to incorrect classifications.

## 4 Discussion

The effectiveness of the ASIF procedure raises questions on the role of memory and retrieval in machine learning, while at the same time opens new opportunities for products based on multimodal models, opening many avenues for future works. In the following we will discuss these aspects.

**Perception and interpretation disentangled.** In ASIF there is no trace of the multimodal data in the weights of the encoders, which are pretrained on different unimodal datasets. Nonetheless, the relative representations and the outcome of any classification task fundamentally depend on the multimodal dataset. This state of affairs reflects the factorization of perception and interpretation in the two stages constituting an ASIF model; the encoding and the construction of the relative representations. Such factorization is desirable because it relieves the black-box neural encoders from the responsibility of attributing meaning to their inputs, as envisaged by Eco [54, Par. 3.3.1.3]. Therefore, we can consider the neural encoders as mere sensors and focus only on the second stage to analyze, explain, and act on the interpretations of an ASIF model, as shown in Figure 6.

**Learning or retrieval?** As we have seen, the ASIF procedure requires no training: it does not distill the multimodal data into any learnable parameter. Rather, it prescribes a rigid memorization of the embeddings of the multimodal dataset, where each entry has its fixed-size spot, similarly to a retrieval process. On the other hand it seems impossible to not describe ASIF as a learning algorithm; for instance it satisfies the fundamental characterization given by Mitchell [55]: the more the multimodal data the more ASIF improves, as we can clearly see in Figure 5. Ultimately, an ASIF model is functionally comparable to CLIP. ASIF blurs the border between learning and retrieval by questioning the effectiveness of storing information only in the weights, and rather advocates to combine learning representations with external memories. We encourage more work on memory augmented neural networks and towards understanding the implications of memory for generalization.

**Generalization to new distributions.** The empirical performance of ASIF calls for a discussion on zero-shot and out-of-distribution generalization in foundation models trained with huge data sets. Clearly, the performance of ASIF will depend strongly on the multimodal data used for alignment. As an example, the good performance on Imagenet may not be particularly surprising in light of the qualitative evaluation seen in Figure 6. There, our query image might as well had been part of our multimodal data set, as the semantic gap with its neighbours appears to be small. Despite this, our choice of downstream evaluation and pre-training data set is identical compared to prior work [2]. As such, while it appears clear that ASIF should fail when the semantic gap between downstream task and "training data" is large, it is unclear why it should be different for more standard models like CLIP [1] and LiT [2]: if a gap does exist, future work should work to address it. In the meanwhile, we recommend that future foundation models are benchmarked on significantly broader sets of downstream tasks, ideally with some analysis of the semantic distance between test and training data (if any). Alternatively, our results may be interpreted in light of the strong performance of unimodal models. There may be a limited benefit of training from scratch on less curated multimodal data sets compared to standard and well established unimodal data sets, although we posit that at even larger scales the difference may be more significant.

**Limitations.** The simple ASIF procedure presented here offers a strong baseline for multimodal models, but its performance still falls apart to CLIP and LiT when the mutimodal dataset is abundant

and the cost of training is not a concern. Additionally, the large dimensionality of the relative representations, even if sparse, poses challenges for directly applying ASIF to tasks like text-to-image generation. We recognize that the experiments reported in this paper do not provide a comprehensive examination of the myriad of downstream tasks multimodal models are known to adeptly handle, it is important to note that such broad coverage was explicitly out of scope for this work. Our primary objective here was to introduce and justify the ASIF procedure, illustrating its effectiveness on the representative task of zero-shot classification. In making this choice we followed [2], that used the very same datasets to showcase the benefits of a locked image encoder, that is their main claim. We anticipate and welcome a more extensive evaluation of ASIF in the context of a wider range of tasks in future research endeavors.

**Conclusions.** We presented a simple procedure called ASIF to assemble a fully functional multi-modal model like CLIP from two unimodal pretrained encoders and a collection of image-text pairs without tuning a single neuron. While achieving competitive zero-shot classification results with its predecessors using a fraction of the data, the proposed ASIF procedure enriches multimodal models with editability–a new model based on different pairs can be deployed in seconds–and interpretable representations. The effectiveness of ASIF models also clashes with the dominant view of a learning algorithm as a way to distill data into the parameters of a model, and raises questions on the role of memory and retrieval in machine learning.

## Acknowledgments

AN, MF, and FL partially worked on ASIF when they were at Amazon Web Services in Tübingen, Germany.

This paper is financially supported by the PRIN 2020 project no.2020TA3K9N (LEGO.AI), PNRR MUR project PE0000013-FAIR, and ERC Grant no.802554 (SPECGEO).

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

# Appendix

In this appendix, we further showcase the interpretability of ASIF models when used for classification in Figure 7. Then we provide additional details for the *scaling laws* and *EuroSAT* experiments presented in the main paper, and report additional results about the impact of the size of the encoders (Table 2), and of the image training dataset. We also report further evidence that the ASIF construction is not overly sensitive to its hyperparameters. Lastly, we discuss more in detail the idea that captions of similar images are alike in Figure 10.

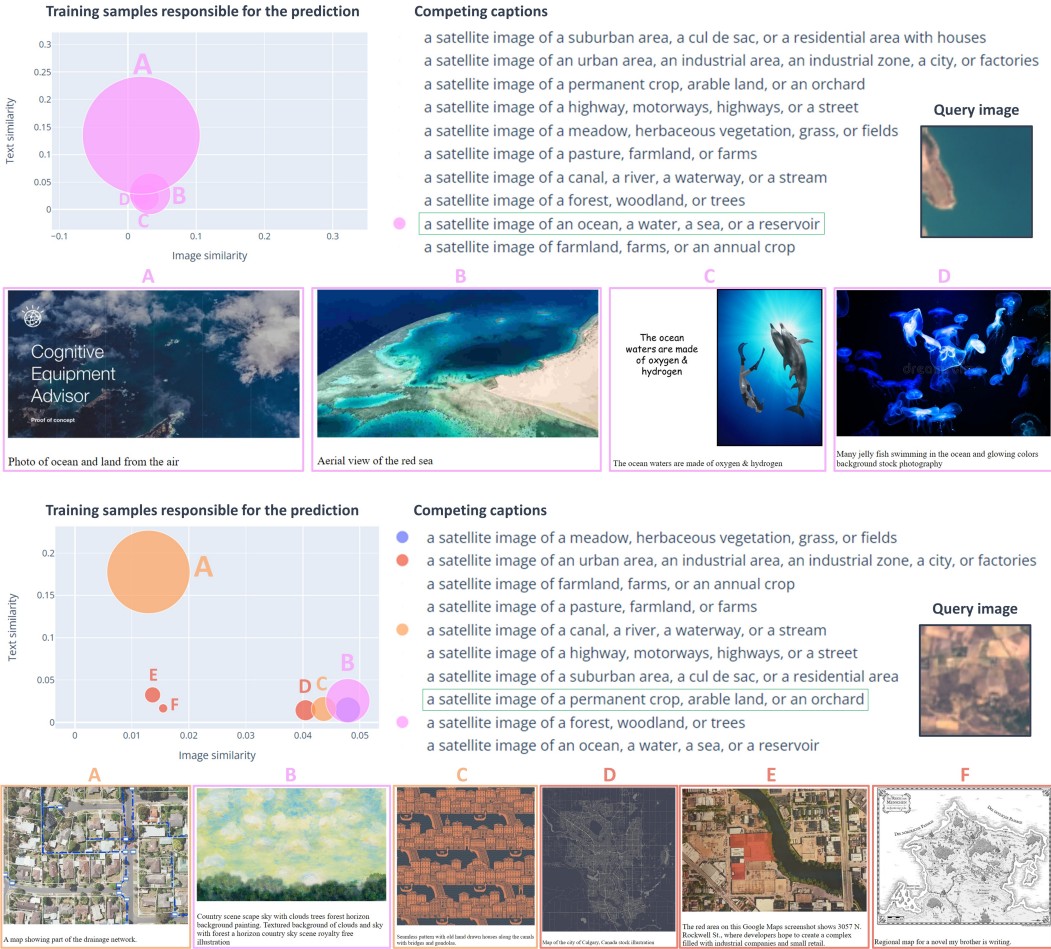

Figure 7: **Interpretability of EuroSAT classifications through ASIF.** Analysis of the classification outcome of two EuroSAT query images using ASIF. The scatter plot shows the samples in the training set closer to the query image and the candidate caption of the corresponding color. Image and text similarity are computed through cosine similarity in the visual space of DINO and the text space of SentenceT. The size of the marks is proportional to the product of the image and text similarity. The class chosen is the one with the largest total area. Below are shown the corresponding pairs from the training dataset CC12M. We can notice the distance between the EuroSAT dataset and the 1.6M samples of CC12M we used, many of the closest images are not from satellite and even then may have misleading descriptions, as image *A* in the second example. An interactive version of this plot for any ASIF classification can be obtained using our code demo attached in the supplementary material.

## A  Additional details on the scaling laws experiment

**Models used in the scaling laws experiments.** As discussed in the main paper, we tested ASIF with smaller image and text encoders to provide early evidence about ASIF scaling laws. We used three different instances of DEIT [44] vision transformers, the tiny (5.6M parameters, 192-dimensional embeddings), small (22M, 384), and base (87M, 768), and the original VITb16 vision transformer [56] (86M, 768). The DEIT models were pre-trained on a smaller dataset, the standard Imagenet1k training set [46], while VITb16 was pretrained on Imagenet21k [47]. As text encoders, we used smaller versions of SentenceT [48], with 23M and 33M parameters (both 384-dimensional embeddings), in contrast to the 110M parameters of the main model (768).

Figure 8 shows that, with smaller encoders producing smaller embedddings, we do not observe a performance saturation within 1.6M image-text couples. Further experiments with larger datasets are left for future work.

**Impact of image pre-training data.** In Table 2 we report the complete results of ASIF models using DEIT encoders [44]. We observe the expected positive correlation between the size of the encoders and the classification accuracy. Interestingly, ASIF with the largest instance of DEIT beats the one based on the standard VIT pretrained on Imagenet21k on three out of four of test datasets, while losing more than 10 points on CIFAR. These results may be interpreted in light of the similarity of the datasets we are using, with features useful to classify CIFAR images less overlapping with Imagenet1k features with respect to the other datasets.

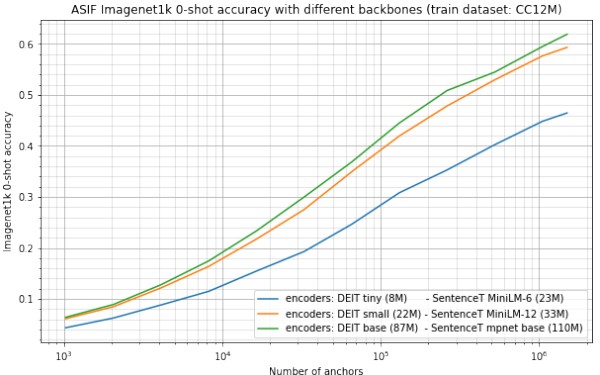

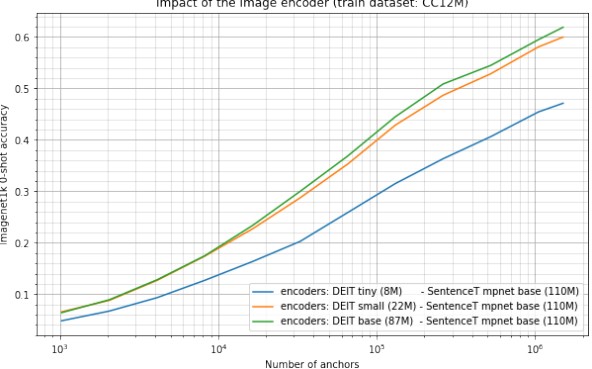

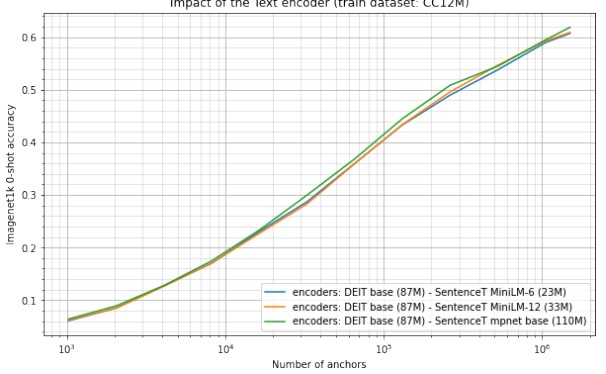

Figure 8: **ASIF performance does not saturate earlier with smaller encoders.** Classification accuracy keeps growing without saturating but is lower for smaller models. Furthermore, we observe that the quality of the vision encoder is more relevant than the quality of the text encoder with respect to zero-shot Imagenet classification.

## B  Additional details on the EuroSAT experiment.

EuroSAT, a renowned benchmark for satellite image classification, serves as a testing ground for out-of-distribution generalization in zero-shot and few-shot scenarios [53]. The dataset contains 27,000 images labeled under ten categories. Our ASIF model with a DINO visual backbone (denoted as 'ASIF unsup' in table 1) achieved a zero-shot classification score of $29.4\%$. While significantly better than random chance, this modest performance is not surprising considering the scarce presence of satellite images in the CC12M dataset.

As a further experiment, we randomly selected 100 images from the EuroSAT dataset and incorporated them into our ASIF training set, raising the total to 1,500,100 image-text pairs and leaving 26,900 images for testing. We created captions for the EuroSAT images using the template "*a satellite image of* [CLASS NAME]". This way the ASIF model improves dramatically, reaching a classification accuracy of $82.5 \pm 2.8\%$ on EuroSAT (average $\pm$ standard deviation of 5 trials).

| ASIF backbones (Params, pre-training data) | ImNet | CIFAR | PETS | ImNet-v2 |
|---|---|---|---|---|
| DEITtiny (5.6M, Im1k) - STminiL6 (23M, see Sec. 3) | 46.5 | 37.3 | 75.6 | 38.3 |
| DEITsmall (22M, Im1k) - STminiL12 (33M, see Sec. 3) | 59.3 | 46.0 | 80.4 | 50.3 |
| DEITbase (87M, Im1k) - STbase (110M, see Sec. 3) | 60.9 | 50.2 | 81.5 | 52.2 |
| VITb16 (86M, Im21k) - STbase (110M, see Sec. 3) | 55.4 | 63.3 | 71.5 | 45.6 |

Table 2: **Zero shot classification accuracy of ASIF models with different backbones**. We observe that the ASIF procedure remains effective even with smaller encoders pre-trained on reduced visual datasets such as Imagenet1k.

Contrarily, CLIP [1], while demonstrating better zero-shot accuracy at $54.1\%$, is trained on a private dataset comprising 400 million images. This dataset may contain a larger number of satellite images than our 1.6 million subset of CC12M. Given the substantial improvement observed when we added just 100 EuroSAT images, it's reasonable to speculate that CLIP's enhanced performance might stem from its larger database of satellite images. However, confirming this theory is impossible due to the private nature of CLIP's training set.

We can, nevertheless, examine the presence of satellite images in the CC12M dataset. Using ASIF models' unique interpretability property, we can trace the training samples behind each classification. Figure 7 displays two EuroSAT samples, one classified correctly and the other not, along with the corresponding CC12M pairs responsible for the classifications. We note that our subset of CC12M is lacking in satellite images, and the few available often have misleading captions, such as a map of a drainage network tagged as "a satellite image of a canal, a river, a waterway, or a stream" instead of an urban area.

The images shown are an adaptation of the interactive plot to analyze any ASIF image classification we provided in the code demo attached in the supplementary material.

## C  ASIF sensibility to its hyperparameters

Finally, we present evidence about the sensitivity of the ASIF model to the hyperparameters $p$ and $k$. Specifically, we show the hyperparameter search for PETS and CIFAR100 in Figure 9. Table C with results on the parameters fine-tuned on the two datasets reveals marginal improvements over the standard choice of k=800 and p=8. This suggests that the ASIF model is relatively insensitive to the choice of these hyperparameters.

| Tuned on | Parameters $p$,$k$ | CIFAR | PETS |
|---|---|---|---|
| PETS | (200,8) | 60.9 | 72.3 |
| CIFAR | (1600,6) | 64.9 | 63 |
| ImageNet1K | (800,8) | 63.3 | 71.5 |

Table 3: Hyperparams search: tuning on each dataset per row.

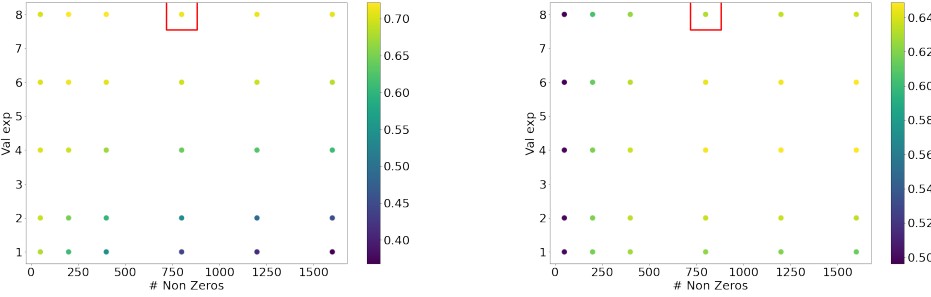

Figure 9: **Hyperparameters search** over Left Pets, Right CIFAR100. Highlighted in the red square the performance achieved tuning on Imagenet1K.

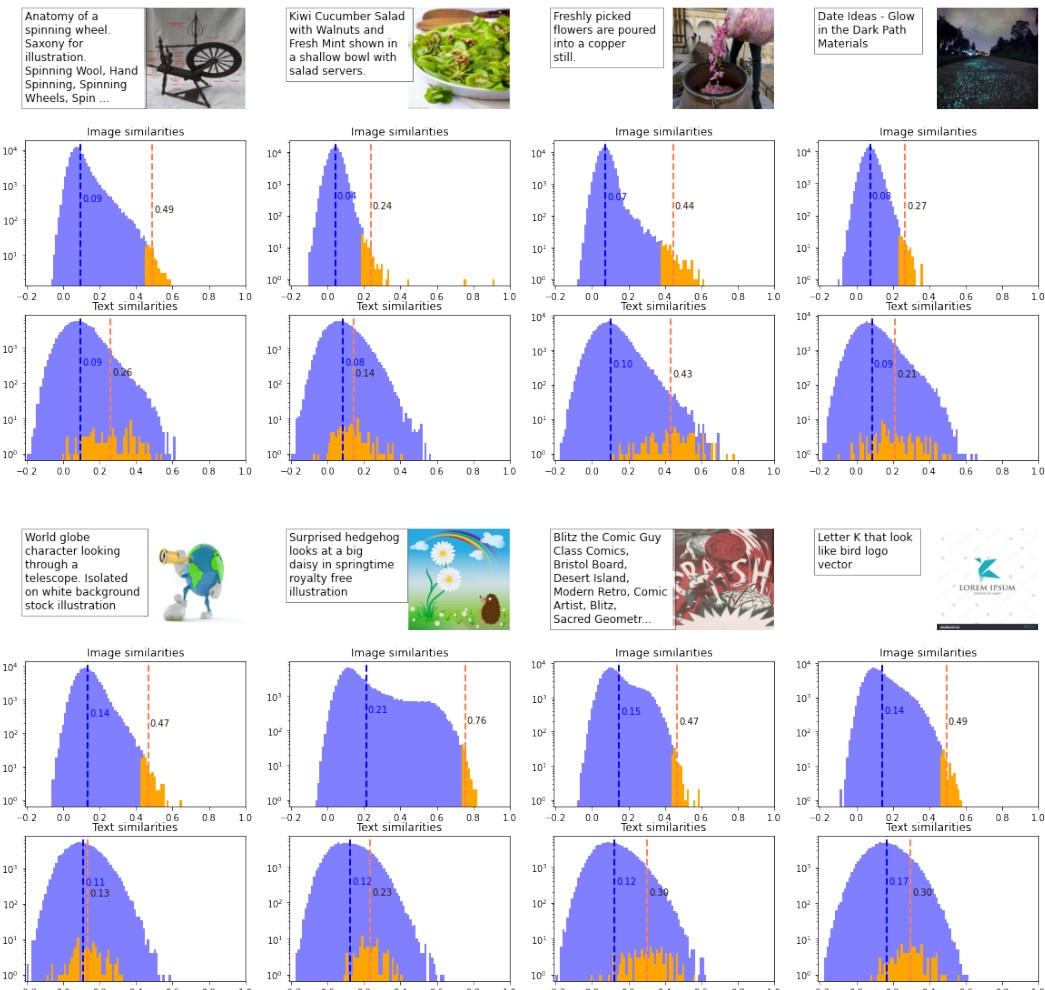

Figure 10: **Caption of similar images are themselves similar.** For 8 image-text pairs, we show in the first row the distribution of the image similarities against $100k$ images in the train set in blue (CC12M), and highlight the 1000 most similar in orange. The dashed lines indicate the mean of the two distributions. In the second row, we show the text similarities against the captions of the same $100k$ (blue) and 1000 (orange) images. If captions of similar images are themselves similar, we expect the dashed orange line in the second row to be at the right of the blue dashed line, as we observe. The average gap between the orange and blue lines in the second row over 10,000 image-text couples from CC12M is $0.098 \pm 0.070$.

