# OpenReview forum: "ASIF: Coupled Data Turns Unimodal Models to Multimodal without Training"
_NeurIPS.cc/2023/Conference — NeurIPS 2023 poster_

### Official Review · Reviewer_QV7o · 2023-07-04

**Soundness:** 3 good
**Presentation:** 3 good
**Contribution:** 3 good
**Rating:** 7
**Confidence:** 3

**Summary:**

This work proposes a method to obtain multimodal representations from pretrained unimodal models and a multimodal dataset, and demonstrates the effectiveness of the method on the zero-shot classification task. The method is to map candidate images and texts into the same representation space, which consists of similarities of the candidates to each image/text in an anchor multimodal dataset.

**Strengths:**

- The proposed method is very simple and intuitive, by representing an image or text as a vector of similarities to an anchor set of images/texts.
- No gradient updates are required. One can directly benefit from pretrained unimodal models. No training parameters are involved.
- The advantage of quick model adjustments via data handling without model retraining.
- The method gives (to a certain extent) explainable multimodal representations.
- The paper is clearly written and easy to understand, including a comprehensive discussion section.



**Weaknesses:**

- A larger multimodal dataset leads to better zero-shot classification performance, but also leads to a bigger representation vector. E.g. a 1.6M-d vector for an image or text, while manageable for retrieval tasks, might be difficult to maneuver for other downstream tasks (e.g. any generative tasks). I understand that the authors stated that effectiveness or applications to tasks other than zero-shot classification is out of the scope of this work, but I'd like to see more discussions on the limitations or impacts of the size of representation vectors.
- The method is sensitive to the curated multimodal dataset. The authors ablated on the size of the dataset, but not the source of the dataset. E.g.what would be the effect of using a different multimodal dataset from CC? (COCO, LAION, etc?)


**Questions:**

As stated in the section above.

**Limitations:**

I don't see potential negative societal impacts of this work.

---

> ### Author Rebuttal · Authors · 2023-08-09
>
> Thanks for the review; we provide some brief comments on the points raised.
>
> “*I'd like to see more discussions on the limitations or impacts of the size of representation vectors.*”
>
> Thanks for pointing this out. Indeed, the large dimensionality prevents a straightforward application of ASIF in tasks like text-to-image generation. Other tasks like captioning could be approached with different pipelines (as described in the answer to 2Qsg). Still, we agree with the reviewer that the higher dimensionality trait of ASIF should be cited in the discussion, and we will add it to the camera-ready version. We also envision follow-up works tackling the problem of reducing the dimensionality of the multimodal space built by ASIF.
>
> “*What would be the effect of using a different multimodal dataset from CC, like LAION or COCO?*”
>
> - For LAION, we expect a similar performance to CC given that they almost follow the same production pipeline (scraping from the internet). LAION would be the natural next step after CC given its larger size.
> - Our very first working demo was built using a subset of COCO, which indeed worked and led us to this work. The problem with COCO is its small size (328k samples) and the relatively narrow distribution of captions: while they are much more precise than CC, they also cover much fewer concepts. CC captions are sometimes imprecise but include different captioning styles and cover a wider set of words. Still, we agree that assessing the performance with the COCO dataset is valuable, and we will add COCO to the training dataset choices in the demo notebook provided in the supplementary material.

---

> > ### Comment · Reviewer_QV7o · 2023-08-20
> > **Response**
> >
> > Thanks to the authors for the rebuttal. While I share the same concern with some other reviewers that the method is unlikely to be widely adopted, I still think the method is novel enough to be published and can inspire and stimulate more ideas along this line. Therefore I maintain my rating of 7.

---

### Official Review · Reviewer_2Qsg · 2023-07-04

**Soundness:** 3 good
**Presentation:** 3 good
**Contribution:** 3 good
**Rating:** 7
**Confidence:** 4

**Summary:**

The paper proposes a method to align vision and language without learning a parametric model. Main idea: having a support set of image-text pairs, the structure of the visual data should match the structure of the language data. More precisely, the distances from one query image to all images in the support set (denoted as relative representations) should be similar to the distances from the associated text and all texts in the support set. For visual tasks that can be formalized as text retrieval, this method can be applied without learning. The method has relatively good performance given its simplicity.

**Strengths:**

Strong points:
 S1. The method is simple and can be easily applied to potentially many vision-language retrieval tasks. Being able to use any independently pre-trained vision and language model is a big advantage. Another big advantage is that it does not require any training.

S2. The model can be easily adapted by simply changing the support set of image-text pairs.

S3. Some interpretability is obtained by inspecting the nearest images and captions used to obtain the relative representations.

**Weaknesses:**

Weak Points:

W1. As the authors notice, the model is slower at training time. Although optimizations might reduce it.

W2. It is not clear if this approach can be used for other downstream tasks, like VQA, or captioning. Although this might be outside of the scope of the paper it might be worth discussing.

W3. The paper will benefit from more comparisons with recent vision-language models. Relevant experiments will be on image-to-text retrieval and text-to-image retrieval on COCO and Flickr30K datasets compared to methods like BLIP [A]

W4. Comparing with some baselines will be beneficial. If we have a support set of the classification task, with ground truth image-label pairs a good baseline is a simple K-nearest neighbor on the same visual representation used in ASIF. For example, using the 100 image-text pairs of EuroSAT we can create a KNN relying on the same image encoder as ASIF. How does this KNN compare with ASIF in terms of performance?

[A] Li, Junnan, et al. "Blip: Bootstrapping language-image pre-training for unified vision-language understanding and generation." ICML, 2022.

**Questions:**

Q1: Can the relative representations be used for other downstream tasks, different than image classification or retrieval?

**Limitations:**

The authors have discussed the limitations regarding inference speed.

There is no specific negative societal impact.

---

> ### Author Rebuttal · Authors · 2023-08-09
>
> Thanks for the review. We appreciate the feedback and would like to offer some brief responses to the points raised.
>
> “*Can this approach can be used for other downstream tasks, like VQA, or captioning?*”
>
> We are indeed conducting preliminary experiments to use ASIF for captioning in a subsequent work. Our current approach is centered on ASIF, combined with an LLM, to generate plausible captions for new images. This process utilizes captions retrieved by ASIF representation and then iteratively refines the crafted caption. As illustrated in Figure 6 of the paper, we expect the LLM to generate expressions such as "a photo of a triumphal arch" based on the retrieved captions and hints at their similarities to the caption sought.
>
> “*More comparisons with BLIP and KNN on EuroSAT*”
>
> - Thanks for pointing this out. We focused on CLIP and LiT since ASIF can be seen as a third step in the research direction traced by LiT. Nevertheless, we urge a comparison with more recent models like BLIP in future works trying to improve ASIF performance using larger datasets and further optimizations. Our primary objective here was to introduce and justify the ASIF procedure, illustrating its effectiveness on the representative task of zero-shot classification. In making this choice, we followed LiT, which used the very same datasets to showcase the benefits of a locked image encoder, that is their main claim.
> - Thanks for the valuable suggestion of making the KNN experiment on EuroSAT, we plan to make the test, expanding the discussion in appendix B. We expect a close but slightly lower performance with respect to ASIF, since ASIF can count on more images useful to perform the classification from its original training set (as seen in Fig. 7 in the Appendix).

---

> > ### Comment · Reviewer_2Qsg · 2023-08-18
> > **Comments after rebuttal**
> >
> > I thank the authors for their rebuttal. Overall I think that this is a good paper, with an interesting, clear idea and good experiments. I will increase my score to 7.

---

### Official Review · Reviewer_UoCi · 2023-07-05

**Soundness:** 2 fair
**Presentation:** 2 fair
**Contribution:** 2 fair
**Rating:** 4
**Confidence:** 4

**Summary:**

This paper proposes ASIF, transferring independently pre-trained image/text encoders to the classification task without further finetuning.  The proposed method only needs a small amount of paired image-text data as anchors, and represent new data samples using the relative representation to the anchor samples. The simple method achieves reasonable results on different image classification benchmarks.

**Strengths:**

1. The proposed approach is efficient since it does not large amount of image and text data pairs to train extra multimodal representations that align images and text. Also, the required image-text pairs can be only 1.6M, which is small comparing with original CLIP and other multimodal classification models.
2. The needed image-text pairs are flexible. By adding more anchor samples for specific image classes, the proposed method could obtain better performances for such specific classes.

**Weaknesses:**

1. Even though ASIF does not need large amount of paired image-text data, it still requires pre-trained single-modal models which are already trained on large amounts of image or text data samples. From this perspective, I didn't see the necessity of such setting.
2. The prediction results are relying on anchor samples. If the anchor samples are not representative enough, the method will not be that useful. Imagine for a target dataset, if the provided anchor samples do not come from the same dataset or do not include similar samples as the target samples, the method may fail. If the authors can provide more experimental results to justify the influence of different chosen anchor samples, it will be more promising.
3. The paper should be re-organized and the writing should be improved a lot. For example, the "ASIF recipe" can be written in an algorithm format, instead of current style.

**Questions:**

1. In Figure 4(b), why the image representation for x* is "absolute representations"? This term is never mentioned in the text.
2. In Figure 4(a), the first sample x1 goes through the image encoder and get the embedding for x1. However, for x2, why such process involves y2 (the black letter y2 under the box for x2)?
In general, all the figures in this work should be re-designed since current presentation is very confusing and difficult to follow. Also, the resolution of figures should be improved.

**Limitations:**

yes

---

> ### Author Rebuttal · Authors · 2023-08-09
>
> Thanks for the review, we hope to have addressed all points:
>
> On the Weaknesses:
>
> 1. “*ASIF still requires pre-trained single-modal models which are already trained on large amounts of image or text data samples.*”
>
> The crucial difference is that unimodal data may come untied and even without any labels, while captioned images are scarcer and more challenging to gather. Furthermore, the ASIF setting may be desirable for two key properties: the ability to easily edit the model and its interpretability.
>
> 2. “*If the anchor samples are not representative enough, the method will not be that useful.*”
>
> We perceive this aspect as a strength rather than a flaw. ASIF deliberately separates perception from interpretation, and only the anchors are amenable for any classification outcome. This way, we can control the visual understanding of the model by curating the limited set of anchors. Effective curation is possible given the swift iterations and the interpretability of ASIF models. And we don't need to imagine; we conducted the experiment suggested by the reviewer! In the EuroSAT experiment, we discuss the impact of adding anchors closer to the distribution of interest (penultimate paragraph of section 3 and appendix B).
>
> 3. “*The writing should be improved a lot*”
>
> We acknowledge this opinion, but the positive feedback we received on the writing, and specifically on the “recipe” format, still largely outweigh the negative feedback. Nevertheless, we want to improve the readability of the figures for the camera ready (especially Figure 4).
>
> ---
>
> On the Questions:
>
> 1. “*"absolute representations" used in Figure 4b but never mentioned in the text.*”
>
> Thanks for bringing this up. We like “absolute representations” because it contrasts with “relative representations,” but indeed, it should be introduced in the manuscript. We will add it in the camera-ready version.
>
> 2. “*Figure 4a not clear*”
>
> We see that this subfigure is prone to misunderstanding. We are considering moving all the captions in line with y1 below. In general, we plan to polish all the figures in the camera-ready version, thanks for this feedback.

---

> > ### Comment · Reviewer_UoCi · 2023-08-19
> > **Response**
> >
> > I would like to thank the authors for clarifying my concerns. Based on the EuroSAT results, ASIF could obtain better performances by adding in-domain examples for training. Also, from Figure 5, the imagenet results increase when adding more image-text pairs. The proposed method looks interesting, as it does not require a large amount of image-text pairs and does not need retraining when the training set is updated. As the results shown in Figure 5 are promising as the results are keeping increasing, but based on current experimental results for ASIF, it is hard to justify ASIF "achieves competitive performance with CLIP and LiT". It will be very interesting to figure out when ASIF could match the performances of CLIP and LiT.

---

### Official Review · Reviewer_BGFe · 2023-07-06

**Soundness:** 3 good
**Presentation:** 3 good
**Contribution:** 4 excellent
**Rating:** 7
**Confidence:** 3

**Summary:**

This paper presents a novel approach to aligning text-image modalities without any training. The method is based on the assumption that images and their captions have similar relative embeddings, even when trained independently. By leveraging paired multimodal data, relative representations can be computed within each modality, and they can serve as a medium for cross-modality communication. The major strength of this approach is its novelty, as it removes the need for training to align different modalities of data.

**Strengths:**

The idea is novel and has the potential to introduce a new paradigm in multimodal alignment research. Adapting to a new distribution of data is easy as it only requires a new set of coupled data without training. The concept of separating perception and interpretation can open up new research opportunities and can be applied to various multimodal problems.

**Weaknesses:**

- Although the approach is interesting, a more comprehensive analysis is needed to convince the readers, including a global analysis of similarities between features from text and image encoders, potentially compared with CLIP and LiT.
- Figure 2 requires additional explanation or a clearer illustration to motivate and convince the reader.
- Figure 4 could be improved for better understanding. The diagram is not intuitive at first, without fully reading and understanding the method section.

**Questions:**

- How can you provide more convincing evidence that the features are communicable across different modalities?
- How sensitive is the selection of text encoders?


**Limitations:**

The method has not been verified across a wide range of multimodal downstream tasks, but it is clearly stated in the limitation section.

---

> ### Author Rebuttal · Authors · 2023-08-09
>
> Thanks for the review, we provide some brief comments on the points raised.
>
> “*More comprehensive analysis of similarities between features from text and image encoders to convince the readers*”
>
> Thanks for the suggestion, in the appendix we conduct an analysis of the two feature spaces that looks at the relative distances (Fig. 7 and 10), we imagine to give it more space when presenting the work, in a future poster or blog post. Maybe the reviewer is imagining a different analysis and comparison with CLIP and LiT, we would really appreciate an expansion on this point in the discussion period, to us is not obvious how to do this given e.g. the different dimensionality of the feature spaces.
>
> “*How can you provide more convincing evidence that the features are communicable across different modalities?
> How sensitive is the selection of text encoders?*”
>
> In this work we demonstrated this effective communication through zero-shot and few-shot visual benchmarks as well as discussing ASIF performance in the audio domain. Still, we acknowledge that further evidence would be beneficial to the adoption of the method. In this work, we focused on introducing the new ASIF methodology but we expect to have further evidence as well as a deeper analysis on the components –including the text encoder– in future work. We also point the reviewer to [1] for a further discussion of the communication allowed by relative representations.
>
> [1] Moschella, Luca, et al. "Relative representations enable zero-shot latent space communication." published at ICLR 2023.

---

> > ### Comment · Reviewer_BGFe · 2023-08-16
> > **Response to Authors' Rebuttal**
> >
> > Thank you for the authors' response. Below are the clarifications.
> >
> > *More comprehensive analysis of similarities between features from text and image encoders to convince the readers*
> >
> > If I understand correctly, Figures 3 and 10 show that when we pick similar images to the top image (using an image encoder), their corresponding captions tend to be more similar to the caption of the top image (using a text encoder). What I was wondering is how similar the new representations from ASIF are compared to those from CLIP and LiT. I understand that ASIF is an unsupervised method so it might not perform better than CLIP and LiT, but I assume that there can be some better cases too.
> >
> > *How can you provide more convincing evidence that the features are communicable across different modalities? How sensitive is the selection of text encoders?*
> >
> > It may be a little bit early to conclude that the current method works well across various text encoders (e.g., naive BERT, naive RoBERTa, and SimCSE). Have the authors considered adopting other text encoders? Given that the primary modalities this paper focuses on are text and vision, discussing this could be an important aspect of the paper.
> >
> > Another concern I have is with the current title of the paper "Coupled Data Turns Unimodal Models to Multimodal without Training." The counterpart models like CLIP and LiT have titles: "Learning transferable visual models from natural language supervision" and "Lit: Zero-shot transfer with locked-image text tuning." Both of these titles clearly indicate the modalities they address. Although the authors include the audio result in the appendix, the primary modalities discussed throughout the paper are text and vision. It seems to me that the audio result is a separate work and more detailed discussion should be open to the reviewers to fully convince ASIF is worth having the current title. I would like to hear the authors' opinion on this as well.

---

> > > ### Author Response · Authors · 2023-08-18
> > >
> > > Thanks for the clarifications, below are our responses.
> > >
> > > *More comprehensive analysis of similarities between features from text and image encoders to convince the readers.*
> > >
> > > Thanks for elaborating on this point. If we have understood correctly, this suggestion could manifest as a qualitative comparison between the inter-similarities of a set of images and texts using ASIF, CLIP and LiT. This would give some indications on the behavior of these models, e.g. we could observe which one between ASIF or LiT is more similar to CLIP. This comparison is very straightforward, and we plan to include it in the appendix of the camera-ready version.
> > >
> > > *Adopting other text encoders and title of the paper.*
> > >
> > > We privileged the variability on vision encoders given the broader diversity of training methods on the vision side, and the well-established position of SentenceT as the strandard sentence encoder in literature.
> > >
> > > We believe that the current title well communicates the main message of our work using a synecdoche commonly used in this research area to represent the broader scope with specific parts. Currently, images and text represent the main domains where multimodal models are tested given the wider availability of relevant models and paired data. Our title would not be an exception in this field, e.g. “Multimodal Neurons in Artificial Neural Networks” by Goh et al. (2021) covers only image and text domains.

---

### Official Review · Reviewer_t279 · 2023-07-27

**Soundness:** 3 good
**Presentation:** 3 good
**Contribution:** 1 poor
**Rating:** 4
**Confidence:** 5

**Summary:**

This paper proposes to leverage single-modal pre-trained text & image encoders and a relatively small image-text dataset to create a CLIP-like open-vocabulary visual recognition model without training. The authors claimed that the proposed model ASIF is more training efficient, more interpretable, and can easier handle data copyright issues than CLIP and alternatives such as LiT. More specifically, relative representations w.r.t. the available image-text pairs are computed for image and text respectively. To remove noise, only top-k values are kept. The resulting sparse embeddings can be used for image classification tasks in a way similar to CLIP. The above process requires no training, and the anchoring image-text pairs can be removed without re-training. Moreover, by inspecting the top-K anchoring image-text pairs, human can understand how ASIF makes its predictions.

**Strengths:**

S1. Using relative representation to create a CLIP-like open vocabulary visual recognition model is new.

**Weaknesses:**

W1. Empirical results are weak.

W2. The major claims (efficient training, data copyright, interpretability) have some flaws.

W3. Scalability / generalization (to more tasks) remains unclear.

The only empirical results this paper presents are image classifications in 4 benchmarks, and the accuracy are far below widely used visual recognition models. I understand given that no training is required, this can still be considered strong results. However, this also means that this method is unlikely to be widely adopted. Moreover, for ASIF with supervised visual encoder (DEIT base), the ImageNet accuracy is also considerably lower than the original DEIT results. This raises a concern that the assumption of a available supervised encoder might not be realistic. Given the 1.6M anchor data, I also suggest the authors to run a baseline of taking existing single-modal encoders (unsupervised) and finetune on these image-text pairs with contrastive loss, so that readers can better understand the trade-offs of ASIF and training.

I also have concerns on the claims. The uni-modal encoder also requires training, and it seems that the text encoder used was trained on 1B internet text, which is much larger than CLIP's 400M. Moreover, if the pre-trained encoders have data copyright issues, there's no way to remove them under the current ASIF framework without re-training. For the interpretability claim, the same procedure could be done for CLIP-like models given an anchoring dataset. (Retrieve nearest neighbors using CLIP's text/image embeddings in the image-text pairs.) Therefore, the interpretability is not ASIF's exclusive advantage.

I acknowledge that the scalability limitation has been properly addressed by the authors. Although some possible techniques to improve it are discussed (L154-L169), I personally don't think this can replace a well-tuned CLIP model. For many real-world applications, inference cost may be more crucial than training cost. On the other hand, the current ASIF is only tested on image classification. I would suggest the authors explore different tasks for future version. Instead of arguing the potential by pure hypothesis, it would probably be more convincing to demonstrate the potential on a wide range of tasks / modality, given that current image classification accuracy are not promising.

Additional Suggestions:
- To make ASIF look more promising, I'd suggest the authors try to utilize relative representations to improve CLIP. In this way, users can choose between finetuning / ASIF depending on the training / inference cost trade-off while getting strong accuracy.
- From the accuracy difference of sup vs unsup encoders, it seems the data quality of pretrained encoders also plays a major role. Studying training data quality vs anchoring data quality might also be an interesting direction.


---------------------------------
Update after rebuttal
================

I appreciate the detailed answers from the authors. Most of my questions are answered and I have a better understanding of this submission; hence the increased confidence from 4 to 5. W2 was properly addressed and the authors promised to clarify the claims in revision. Unfortunately, W1 and W3 remains concerns (no additional results are provided) that outweigh the strengths. I've increased overall rating from 3 to 4 but still leaning negative.

**Questions:**

My questions and suggestions are included in the weaknesses section.

**Limitations:**

Limitations are properly addressed.

---

> ### Author Rebuttal · Authors · 2023-08-09
>
> We invite further dialogue with the reviewer in the next phase to foster an updated assessment of our research that builds upon the discourse. We also thank the reviewer for the interesting points regarding the interpretability and scalability/inference tradeoffs
>
> "*The only empirical results this paper presents are image classifications in 4 benchmarks” and “lack of evidence” on other tasks*
> - The primary evaluation follows the protocol of LiT (table 2 of LiT paper) and employs exactly the same four classification benchmarks used in ASIF.
> - In addition, we also present performance results on EuroSAT (as indicated in the penultimate paragraph of the Empirical Evidence section and detailed in appendix B) and we discuss ASIF's performance on audio data from published follow-up work (appendix C).
>
> *"ASIF with supervised visual encoder (DEIT base), the ImageNet accuracy is also considerably lower than the original DEIT results. This raises a concern that the assumption of a available supervised encoder might not be realistic"*
>
> A comparison against supervised learning on ImageNet is unfair as we focus on open vocabulary classification (also LiT ImageNet accuracy would be considerably lower). Similarly, it would be unfair to report performance of DEIT on any other data set without finetuning. The trade-off between specialization and generalization in image classification is well-known and has been widely discussed in recent literature, including multimodal open-ended models [1, 2, 3].
>
> [1] Wortsman, Mitchell, et al. "Robust fine-tuning of zero-shot models." CVPR 2022.
>
> [2] Wortsman, Mitchell, et al. "Model soups: averaging weights of multiple fine-tuned models improves accuracy without increasing inference time." ICML 2022.
>
> [3] Radford, Alec, et al. "Learning transferable visual models from natural language supervision." ICML 2021.
>
> "*uni-modal encoders also require training, and it seems that the text encoder used was trained on 1B internet text, which is much larger than CLIP's 400M*"
>
> It is essential to clarify that we are not comparing like-for-like here. It is important to distinguish free-form text, which lacks labels, from a collection of captioned images, which are comparatively more scarce and challenging to gather. What is intriguing is the modest quantity of captioned images that ASIF relies upon (1.6M) after the encoders are pre-trained from purely unpaired and potentially unlabeled data.
>
> "*Interpretability*"
>
> Our approach is different because the prediction for us is mechanicistically caused by the nearest neighbour, if you were to change the anchor set then the prediction would change and not just the explanation. There is a vast literature on explainability, arguing that explanations need to faithfully summarise the internal model computations, see [1]. In our case, the interpretability pertains to the assignment of captions to visual features.
>
> [1] B-cos Networks: Alignment is All We Need for Interpretability, Bohle et al., CVPR 2022
>
> “*Trade-offs between training and inference costs*”
>
> The reviewer is correct that inference cost is usually a blocker for industrial applications of foundation models, as large-scale deployment can eclipse even the high training costs. Nevertheless, we contend that the training cost remains a relevant consideration, and the additional attributes of ASIF can potentially position it as the preferred model.
> The idea the reviewer suggest of merging ASIF with CLIP is very interesting. While it is beyond the scope for this submission, being able to navigate the training/inference cost trade-off is very valuable depending on the application. We will be happy to add a discussion in the paper about the inference cost vs training trade-offs and how ASIF may be an extreme point with valuable in-betweens. As it is not obvious how to do this, we would leave this open to future research and acknowledge the reviewer contribution to the discussion.
>
> “*Studying training data quality vs anchoring data quality might also be an interesting direction.*”
>
> We agree with the reviewer and have already begun investigating a closely related avenue for our future research. Specifically, we are delving into the realm of degrading the quality of encoders within an ASIF model while maintaining acceptable performance levels. One possibility is to modulate the quality of the encoders by acting on the training dataset. We thank the reviewer for this valuable suggestion.

---

> > ### Comment · Reviewer_t279 · 2023-08-14
> > **Response to Authors' Rebuttal**
> >
> > I appreciate the authors' response. However, many of my concerns are not resolved, so I don't plan to change my final rating at this point.
> >
> > Regarding W1, I still think the empirical results are way too weak. One of my original point was "the accuracies are far below widely used visual recognition models". **Given the current experimental results, I don't believe building upon ASIF can lead to satisfactory image classification model if we care about the accuracy performance.** Regarding the authors' rebuttal, I don't think it's changing my opinion about this submission but I'll still respond here. I think benchmarks in LiT's Table 1 can also be considered. Table 2 seems merely a small scale ablation for LiT. I admit I missed the EuroSAT results in the supplementary, but my concerns are still unresolved. Did ASIF achieves comparable accuracy with other open-vocab models such as CLIP/LiT on EuroSAT? The table is mainly ablating ASIF design choices only. Lastly, I'm not sure if we can include Appendix C in the discussed here, since it's not this submission's contribution. Adoption of ASIF by others might be a positive signal, but I'm not sure how to evaluate this under the current reviewing process.
> >
> > Perhaps I wasn't clear enough in my original review on the DEIT-related concern. I'm not asking to surpass a specific model's accuracy. I'm questioning the assumption of an existing supervised visual encoder. ASIF significantly hurts the supervised encoder's performance on the same image domain (ImageNet). Empirically, this may indicate that ASIF is not a good method to convert DEIT to open-vocab within a similar image domain. If an supervised pre-trained model is assumed, I would suggest to demonstrate ASIF's advantage in a domain transfer settings.
> >
> > With W2, I would like to emphasize my concerns on the "data copyright issues". This paper claims that "an asset owner can remove their
> > 39 data from the model" (L38) and "... if we need to remove samples ... lost the license to use them ..." (L286-288). If the data to remove is used when training the encoders, I don't think ASIF can handle it very well. **This false claim remains a significant issue for this submission.** To solve this, the encoders need to be retrained, and the 1B text data would become a problem. The cost could be as significant as re-training a CLIP model (400M image-text). The author's rebuttal focused on the data collection aspect, but I'm more concerned on the training cost considering the data removal claim.
> >
> > The interpretability response is confusing. Why do we need to change the anchor set? For the CLIP model, if a set of image-text pairs from its pre-training data is sampled, isn't it a good interpretable anchor set as in ASIF? I still don't understand how ASIF's interpretability is special.
> >
> > For W3 the scalability concern, I'm suggesting making this paper a study of training / inference trade-off. Perhaps ASIF can be done together with the CLIP framework: Pre-train on smaller dataset to save training cost, and add ASIF later. On the other hand, use smaller ASIF anchor set but pre-train on more CLIP data to save inference cost while maintaining reasonable accuracy. The above is just one possibility. To conclude, **I strongly suggest to shift the direction to make ASIF compatible with existing CLIP-like model for a stronger accuracy performance and better scalability**, and show that ASIF can enable useful trade-offs on certain condition / assumption. To be honest, the current scope of this submission is not interesting, at least to me.
> >
> > Overall, my main complaint about this submission is that, being an empirical paper, the experimental results are not convincing to change the community's wide adoption of CLIP. Unless ASIF can work well together with other existing methods and achieve reasonably good accuracy (in 2023 standard), I am unlikely to change my rating. I agree with the other reviewer's positive view about this paper -- relative representation to make open-vocab model is interesting. However, this idea by itself, without a good execution to produce a model that actually works well, does not have enough contribution to the community in my opinion.

---

> > > ### Author Response · Authors · 2023-08-15
> > >
> > > While we respect the Reviewer's opinion, and acknowledge the significant effort in pointing out what they perceive as major weaknesses in our work, we strongly disagree.
> > >
> > > In the interest of a positive and balanced discussion, we would like to stress the following points:
> > >
> > > ### Goal of the paper
> > >
> > > One can turn a pre-trained unimodal model into an open vocabulary one without any further training. This finding is surprising and useful. At the same time, we never advocated ASIF as a one-stop replacement for CLIP or LiT. See ad verbatim block (L359)
> > >
> > > > *The simple ASIF procedure presented here offers a strong baseline for multimodal models, but its performance still falls apart to CLIP and LiT*
> > >
> > > ### Data copyright issue (editability property of ASIF).
> > >
> > > We believe in faith this to be just a tragic misunderstanding that is influencing negatively the judgment of this reviewer: the reviewer states that
> > >
> > > >*This paper claims that "an asset owner can remove their 39 data from the model"*.
> > >
> > > Here is the complete sentence from the introduction, covering the general problem:
> > >
> > > > *Still, training neural networks at such scale presents several challenges beside the obvious infrastructure and training costs. Notably, it requires collecting massive training sets, making it difficult to interpret the predictions of the model in light of their training data. Additionally, the training assets are often not owned by the institution training the model [6]. This introduces several additional challenges, from reproducibility to the difficulty of ensuring that **an asset owner can remove their data from the model [7--11]***.
> > >
> > > The editability property is specific to the multimodal data set. ASIF models can be adjusted in seconds, by easily adding new multimodal samples or removing them (e.g. for copyright issues) by simply adding or deleting their embeddings. It is clear that this capability does not hold for the pretrained encoders. If needed, we will make it clearer in the camera ready.
> > >
> > > Finally, we believe that large scale pre-training on established data sets can be both effective (see imagenet pre-training) and safer from copyright issues compared to downstream data that did not go through the same level of scrutiny.
> > >
> > > See ad-verbatim quotes (L136)
> > >
> > > > *In our procedure, the encoders can be pre-trained with established data sets that do not change over time, and removing the effect of a multi-modal example is as simple as deleting its embeddings.*
> > >
> > > ### Interpretability and relation with CLIP
> > >
> > > > *I'm suggesting making this paper a study of training / inference trade-off. Perhaps ASIF can be done together with the CLIP framework: Pre-train on smaller dataset to save training cost, and add ASIF later.*
> > >
> > > While we found it interesting, note that this is not the goal of the paper. As written above, we are showing how to turn frozen unimodal models into open vocabulary. This comes with additional nice properties. Whether they can be applied to CLIP or not it's besides the point.
> > >
> > > As we already explained in the rebuttal, ASIF cannot be used to explain CLIP predictions as CLIP is not necessarily relying on those samples in a mechanicistic way to make predictions. Clearly, the explainability procedure only relates to the multi-modal data, not the pre-training.
> > >
> > > > *Overall, my main complaint about this submission is that, being an empirical paper, the experimental results are not convincing to change the community's wide adoption of CLIP.*
> > >
> > > We never claimed this to be a purely empirical evaluation of CLIP like models. We proposed a new method with the goal repeatedly stated in this response.
> > >
> > > ### Supervised encoders
> > >
> > > ASIF does not assume to have a supervised visual encoder, it can also be unsupervised as demonstrated. We are interested in evidencing that ASIF works with potentially any uni-modal encoder, either trained through a supervised or unsupervised task. Note that also LiT hurts the supervised encoder's performance on imagenet as is for us with DEIT. At the same time, one gains the open vocabulary capabilities.

---

> > > > ### Comment · Reviewer_t279 · 2023-08-16
> > > >
> > > > Thanks for the response. I'll reply each point in details below.
> > > >
> > > > I think my understanding of the scope and goal of this submission is roughly on the same page with the authors. However, I simply think this scope is not exciting, at least to me. Regarding the goal to convert supervised encoder to open-vocab encoder, isn't there many possible ways that may achieve much better accuracy (LiT is one example)? New ideas are definitely welcomed, but in my opinion, a good execution that demonstrate the idea "works" (not necessarily SOTA but at least compatible with existing strong model) is required for an empirical paper to make actual contributions. LiT obviously works pretty well in practice and achieves the same goal. I'm not sure I can say ASIF works in practice. It is still at an "interesting idea" stage. The above is why I suggest to incorporate ASIF with CLIP in my review/response to improve this work on the empirical usefulness side. I would strongly suggest the authors drop the "completely no training" setting, since it seems unlikely this setting would achieve something useful in the near future.
> > > >
> > > > If the "data removal" difficulty of prior work is not solved by ASIF, perhaps it should not be mentioned. I was misled by this sentence, thinking that ASIF can solve this issue but later being disappointed to find out this is still a limitation. The multi-modal anchoring set is proposed by the authors. Although the ASIF-specific part does not worsen this issue, ASIF does not solve this existing problem as well. May I conclude that ASIF does not make contribution in the "data removability" aspect?
> > > >
> > > > I'm not a legal expert so I can't determine whether public research datasets can be used in commercial products without copyright concerns. Therefore, I can't make a conclusion that ImageNet is 'safer' than web image-text pairs.
> > > >
> > > > The interpretability claim is still not very clear to me. Could you elaborate more in details? We we extract a small image-text pairs set from CLIP's pre-training data for CLIP's interpretability, how's this conceptually different from ASIF's anchoring set for ASIF's interpretability? Also a more higher level question, how is ASIF's interpretability better than the CLIP interpretability hypothesis I mentioned?
> > > >
> > > > Coming back to the goal of ASIF (supervised->open-vocab model w/o training), if ASIF has none of inference time, data removal, or interpretability advantage over CLIP, how is ASIF useful in any way? Users can just use CLIP model (and train on safe paired image-text data) to achieve open-vocabulary image classification need, like LiT did. Restricting to no-training doesn't make much sense here. Hence, I strongly suggest the authors study both no-training and small-training settings.
> > > >
> > > > For the supervised-encoder discussion, I think I understand the authors' claim now. I just think that it's meaningless to evaluate something that's already trained on ImageNet supervisedly  in 'open-vocab' setting again on ImageNet. Perhaps there's a better way to present the table.
> > > >
> > > > Overall, as a reviewer, I'm evaluating both the novelty and **usefulness** of this submission. This submission definitely has values on the novelty side, as recognized by several other reviewers. However, I'm questioning the usefulness, and I think this aspect gets even more important with empirical papers (relative representations are proposed by others, so I consider this submission applying existing method to new problems, which is quite empirical). I agree most of the claims in the manuscript are sound and the presentation is mostly clear, as indicated in my original review's categorical ratings. Nevertheless, are we salified with a "safe" paper with limited evaluation under a limited setting? My answer is no. From my experience in publishing and reviewing NeurIPS papers, I think at least evaluating on a small training set to see if ASIF can help existing open-vocab model is a very reasonable request (might be difficult to finish in rebuttal, but is a reasonable exp to include at submission time). My overall decision is based on the overall rating 3's "weak evaluation" description, thus I still don't plan to increase the score for now. I'm open to further discussions with the authors and fellow reviewers.

---

> > > > > ### Author Response · Authors · 2023-08-18
> > > > >
> > > > > It is clear we disagree with the reviewer. While we will take the feedback into account for future work, it is not reasonable to ask to turn this project into something else more appeasing to them.
> > > > >
> > > > > We will now only correct the incorrect statements of the reviewer.
> > > > >
> > > > > -   "*isn't there many possible ways to convert supervised encoder to open-vocab encoder (LiT e.g.)?*"
> > > > >
> > > > >     Currently there is no way to easily convert a supervised encoder to an open-vocab model except for ASIF. LiT requires to train the whole text encoder. Why this matters? See below the point on usefulness.
> > > > >
> > > > > -   "*a good execution that demonstrate the idea "works" (not necessarily SOTA but at least compatible with existing strong model) is required for an empirical paper to make actual contributions.*"
> > > > >
> > > > >     Results in table 1 demonstrate that ASIF performance is compatible with existing strong models. For instance ASIF overperforms in every benchmark the CLIP implementation made by LiT authors, used as the main evidence supporting the LiT method over CLIP in the LiT paper (table 2 of the LiT paper).
> > > > >
> > > > >     "*I'm not sure I can say ASIF works in practice.*"
> > > > >
> > > > >     The reviewer can easily reproduce our results with the code we provided, there is no training needed.
> > > > >
> > > > > -   "*the "data removal" difficulty of prior work is not solved by ASIF*".
> > > > >
> > > > >     The "data removal" difficulty of prior work is partially solved by ASIF: multimodal data can be easily removed or added without any retraining. The difficulty remains for the training data of the encoders.
> > > > >
> > > > >     We emphasize that even a partial solution for multimodal data is significant. For instance, suppose some pairs in the multimodal dataset are discovered to be flawed: e.g., the image does not correspond to the text, or the text's association with the images raises ethical concerns. It is important to note that such problems, unique to multimodal data, do not arise in unsupervised models like DINO, that we used as ASIF encoder. In such a scenario, a CLIP or LiT model would necessitate a complete retraining, whereas an ASIF model could be adjusted within seconds.
> > > > >
> > > > > -   "*How is ASIF's interpretability better than the CLIP interpretability hypothesis I mentioned*?"
> > > > >
> > > > >     The CLIP inrepretability hypothesis mentioned by the reviewer prescribes to use an ASIF model with CLIP encoders. Therefore we do not see any concern here.
> > > > >
> > > > >     Nevertheless, the interpretability of an ASIF model with unsupervised encoders would still be superior to the above solution since the only link between images and textual concepts would be provided by the anchor set, while in the first case could come from the embeddings, therefore the responsibility of a wrong association could not be ascribed uniquely to the subset of anchors selected.
> > > > >
> > > > > -   "*how is ASIF useful in any way?*"
> > > > >
> > > > >     While we do not believe usefulness to be an imperative goal of a scientific submission to a scientific conference, ASIF does indeed excel in this aspect.
> > > > >
> > > > >     ASIF is useful in settings with scarce training resources or scarce multimodal data, and when interpretability of classification results or fast editability of the model (addition or removal of multimodal training data) is needed. Moreover we believe ASIF can act as a strong and simple baseline for future multimodal work (as also evidenced by the audio results).
> > > > >
> > > > > -   "*I just think that it's meaningless to evaluate something that's already trained on ImageNet supervisedly in 'open-vocab' setting again on ImageNet.*"
> > > > >
> > > > >     Our choice of downstream evaluation and pre-training data set is identical compared to prior work in the same area (LiT vision encoder is also trained supervisedly on ImageNet and evaluated in 'open-vocab' setting again on ImageNet). Still, we share this concern with the reviewer and indeed raise it in the paragraph "Generalization to new distributions" in the Discussion section.

---

> > > > > > ### Comment · Reviewer_t279 · 2023-08-18
> > > > > >
> > > > > > Thanks for the response. I don't have other correctness concerns for this submission, given that data removal claim will be toned down in the revision. I also recognize that the interpretability method is one contribution of this paper, and not limited to ASIF itself. Additionally, I understand that "no training" itself could be very useful for under-resourced audiences. Therefore, I'll slightly increase my ratings from 3 to 4.
> > > > > >
> > > > > > However, I still think experiments demonstrating that some small training could further help ASIF would greatly adds to the significance of this submission, given that GPU compute has become much more accessible (e.g. Colab, Kaggle). Broadening the scope of this project to consider both "no-training" and "small-training" would greatly benefit the community and reach more audiences. ASIF could become a complimentary approach to LiT's "heavy training", and perhaps create a unified framework for users to trade-off training / inference / data resources. From the above discussion threads, it seems the authors are not willing to improve the submission's scope at all, and not even one small additional experiment was done to demonstrate new possibilities during this rebuttal & discussion period.
> > > > > >
> > > > > > Given the limited scope, I don't think rejecting this paper will hurt the community. On the other hand, accepting this paper also probably wouldn't hurt, as this paper is mostly correct and the overall goal has never been done before. Therefore, my overall rating is borderline leaning negative, given the weak empirical contribution (usefulness), and the unlikeliness of the authors improving it if we accept at its current scope.

---

### Author Rebuttal · Authors · 2023-08-10

We thank the reviewers for showing keen interest in our ideas, and for their thorough and quite valuable comments. This general response summarizes the main points raised and how we have addressed them. Specific answers are then provided to each reviewer in response to their remarks.

1.  **Empirical Results and Benchmarks:** The concern regarding the limited empirical results was addressed by clarifying that our evaluation followed established protocols (same as LiT) and included additional results on EuroSAT. We also expanded on ASIF's performance on audio data.

2.  **Comparison with Supervised Encoders and ImageNet Accuracy:** We explained that our focus on open vocabulary classification makes direct comparisons with models like DEIT unfair. We emphasized the specialization and generalization trade-off in image classification, supported by recent literature.

3.  **Training Data of Encoders:** We clarified the concerns about the large training set of unimodal encoders, remarking that ASIF's distinctive capability is to rely on a modest amount of captioned images, which are more scarce and challenging to collect than unpaired and unlabeled data.

4.  **Interpretability and Inference Costs:** We provided insights into our approach to interpretability and responded to concerns about the trade-offs between training and inference costs. A commitment was made to discuss these trade-offs further in the paper.

5.  **Anchor Samples and Model Control:** The importance of representative anchor samples was discussed, highlighting ASIF's deliberate separation of perception from interpretation, allowing for control and interpretability.

6.  **Writing and Figure Clarity:** Acknowledging some concerns about the writing style and figure clarity, we committed to improving these aspects in the camera-ready version of the paper.

7.  **Potential Extensions and Sensitivities:** Responses were provided on the potential use of ASIF for other downstream tasks, sensitivity to text encoders, and comparisons with other models. We highlighted ongoing experiments and plans for future work.

8.  **Representation Vectors and Dataset Choices:** We agreed to discuss the limitations or impacts of the size of representation vectors and the effects of using different multimodal datasets. We also outlined our experience with different datasets and plans to expand on this in supplementary material.

In summary, we believe that we have addressed the main concerns and provided clarification on several key aspects of our methodology. We also appreciated the valuable suggestions for future research and will consider them in ongoing and subsequent work. Thanks once again for the thoughtful engagement with our paper.

---

### Comment · Area_Chair_uPty · 2023-08-13
**Please respond to author rebuttal**

Dear Reviewers,

Please respond to authors after carefully reading the author rebuttals and other reviews. If your assessment of the paper changes, please update your score with a short justification for the new rating.

The paper received diverging initial reviews. Please consider discussing with the authors or other reviewers to see whether we can reach a consensus.

Thank you,
AC

---

### Decision · Program_Chairs · 2023-09-21

**Decision:**

Accept (poster)

**Comment:**

The paper demonstrates aligning multimodal encoders' representation spaces can be done without training using reasonable amount of image text pairs. This is an interesting problem and this paper is the first to show this. The reviewer ratings are 3 accepts and 2 borderline rejects. In the final comments, one rejecting reviewer mentioned the paper's goal is novel and accepting it wouldn't hurt, but is afraid that the authors will not improve the paper further if the paper is accepted. The other rejecting reviewer mentioned the method is interesting and promising but has reservation about the "competitive with CLIP/LiT" performance claims in the paper. The AC agrees that the addressed problem is an interesting and important one and this paper represents an important first step, thus recommends acceptance of the paper. At the same time, the authors are encouraged to address the concerns especially from the two leaning negative reviewers to improve the camera-ready publication.